# Review of Gas Dynamic RF-Only Funnel Technique for Low-Energy and High-Quality Ion Beam Extraction into a Vacuum

**DOI:** 10.3390/mi14091771

**Published:** 2023-09-15

**Authors:** Victor Varentsov

**Affiliations:** Facility for Antiproton and Ion Research in Europe (FAIR), Planckstraße 1, 64291 Darmstadt, Germany; victor.varentsov@fair-center.eu; Tel.: +49-61597-11638

**Keywords:** RFQ and SPIG beam cooler, buffer gas flow, RF-only funnel, RF buncher, gas dynamic and Monte Carlo ion-trajectory simulations

## Abstract

This paper reviews the development and present status of a novel gas dynamic RF-only funnel technique for low-energy ion beam extraction into vacuum. This simple and original technique allows for the production of high-quality continuous and pulsed ion beams in a wide range of masses, which have a very small transverse and longitudinal emittance.

## 1. Introduction

Low-energy radioactive ions extracted into vacuum conditions from gas stopping cells for different investigations, such as high-precision nuclear mass measurements, laser and decay spectroscopy, ion mobility, ion trap techniques and so on, are usually carried out by means of conventional skimmers [1,2,3], radio frequency quadrupole (RFQ) [4,5,6,7,8,9,10,11,12,13,14,15,16,17,18,19,20,21] or radio frequency sextupole (SPIG) [22,23] rod structures.

During the last thirty years, a number of different modifications and constructions of RFQ and SPIG ion beam coolers, as well as the traditional (RF + DC) funnels and carpets (when the DC gradient along the funnel electrodes is additionally used for the ion transport) have been developed. Readers can easily find their descriptions elsewhere, for example, in reviews [24,25] and links within them. Since these conventionally used ion-beam manipulation techniques are not the subject of our present review, we consider it inappropriate to describe them here.

A new approach for ion beam cooling and extraction into a vacuum was proposed in 2001 [26]. It allows for replacing the traditional RFQ or SPIG rod structures or skimmers with a simple and effective gas dynamic RF-only funnel. In this paper, we review the development of the novel gas dynamic RF-only funnel technique, based on this approach [26].

## 2. General Description

The schematic of the new approach [26] for ion beam cooling and extraction into a vacuum is shown in Figure 1.

The primary energetic radioactive ions of interest enter from the left into a gas cell filled with the high-purity buffer gas (usually it is helium or argon) by passing through a thin entrance window, and are stopped there by collisions with the gas atoms. The traditional DC-cage and the (DC + RF) funnel, which are usually installed inside the large gas cells, transport stopped ions to the gas cell exit, where a nozzle with a small throat diameter is placed on the cell axis. Passing through this nozzle, the ions are extracted via supersonic gas jet into the RF-only funnel installed on the nozzle axis in the vacuum chamber, with a background pressure of a few mbar that is maintained by pumping.

The RF-only funnel consists of a stack of thin metal electrodes with decreasing inner diameters towards the exit, separated by gaps for effective gas pumping. RF voltage is applied to the ring electrodes at alternating phases (180° shift) between adjacent electrodes. Such an arrangement creates a repulsive quasi-stationary effective potential in a radial direction that confines ions inside the funnel. Transiting the funnel, the ions experience both the effective potential near the electrodes’ inner surfaces and collisions with a buffer gas. There is no DC gradient along the funnel axis, and the transport of ions through the funnel relies solely on the gas dynamic inside the funnel. Thereby, the construction and RF feed to this funnel are very simple, as only one lead per phase is required to drive the opposite polarities of the electrodes.

We have investigated the operation of the described above RF-only funnel approach using a number of detailed gas dynamic and ion-trajectory Monte Carlo simulations. Gas dynamic simulations of the buffer gas flow were performed using the VARJET code. This code is based on the solution of a full system of time-dependent Navier–Stokes equations for multicomponent gas mixtures and is described in detail in [27]. The results of the gas dynamic simulations (flow fields of the buffer gas velocity, density and temperature) were then used in Monte Carlo ion-trajectory simulations, where the electric fields were simulated with the use of SIMION 8.1 [28].

Moreover, it has been experimentally tested at Stanford University [29,30], ETH, Zürich [31,32], Technical University and Darmstadt [33]; currently, such extraction RF-only funnels are components of the facilities being developed at the Notre Dame University [34,35], McGill University, Montreal [36] and GSI, Darmstadt [37].

## 3. RF-Only Funnels for Ion Extraction into a Vacuum from Gas Cells

### 3.1. The RF-Only Ion Funnel at Stanford University

The apparatus (see Figure 2 in Ref. [30]) with the RF-only ion funnel has been created at Stanford University to investigate the feasibility of extracting Ba^136+^ ions from 10 bar Xe gas into a vacuum. The realization and trial of this RF-only funnel is an important step towards its application in the search for neutrinoless double-beta decay in gaseous ^136^Xe. The identification (Ba tagging) of the ions produced in the double decay of the ^136^Xe isotope would drastically reduce the backgrounds, which are dominated by radioactivity unrelated to the production of Ba ions in the time projection camber (TPC) of the EXO detector.

A schematic view of Stanford’s RF-only funnel, combined with the results of the gas dynamic simulation for the xenon velocity flow field, is shown in Figure 2 [30]. 

The converging–diverging conical nozzle has a subsonic half-angle of 45° and a supersonic half-angle of 26.6°. The nozzle throat diameter is 0.28 mm, the subsonic and supersonic part lengths are 0.5 mm and 15.7 mm, respectively, and the exit diameter of the supersonic nozzle is 16.0 mm. The xenon mass flow rate through the nozzle is equal to 45.3 mbar l/s at a PA of10 bar. 

The RF-only funnel contains a stack of 301 stainless steel electrodes of 0.1 mm thickness with gaps of 0.25 mm between neighboring electrodes. The electrodes are annular with an outer diameter (OD) of 28 mm and an inner diameter (ID) decreasing from 16.0 mm to 1.0 mm in constant steps of 0.05 mm per electrode. The electrodes have three mounting tabs that provide maximal spatial rigidity in the stack and allow for the required tolerances. To illustrate, Figure 3, Figure 4 and Figure 5 show photos of the first (ID 16.0 mm) and last funnel electrodes (ID 1.0 mm), the funnel assembly prior to installation on the nozzle (the view from the base to the exit) and a photo of an assembled RF-only funnel mounted on the nozzle, correspondingly.

A buffer gas differential pumping system can be implemented using RF-only funnels installed in series. This allows for the use of small vacuum pumps to extract ions into a vacuum, and as a result, a significant reduction in the size and cost of the installation. For this purpose, in 2016, we proposed the so-called “triple-funnel system” for the extraction of Ba ions into a vacuum from 10 bar xenon. Figure 6 shows a schematic view of this triple-funnel system combined with the results of a detailed gas dynamic simulation. The detailed information on the nozzle and funnel geometry, gas flow rates through the nozzles and required speeds of vacuum pumps, RF funnels’ operation parameters and so on can be found in Ref. [38].

We proposed another RF-only double-funnel system [39] for ions extraction ions from a cryogenic gas stopping cell for the Super-FRS at FAIR [21].

### 3.2. The RF-Only Ion Funnels for UniCell Setup

The use of the RF-only funnel technique looks promising for experimental study of properties of super heavy elements (SHEs), e.g., a concept of a new Universal High-Density Gas Stopping Cell Setup for study of gas-phase chemistry and nuclear properties of super heavy elements (UniCell), as proposed in [37]. It is under development at GSI, Darmstadt and will be used in experiments with the recoil separator TASCA [40].

The conceptual design variant of the UniCell for the study of SHE nuclear properties is shown schematically in Figure 7. This setup consists of four differentially pumped vacuum chambers connected through three miniature nozzles.

Through the first nozzle, ions are extracted by a supersonic gas jet into the first differential vacuum chamber, where a background pressure of a few mbar is maintained by pumping. Then ions are quickly and efficiently transported by the gas flow through two conical RF-only funnels. These two RF-only funnels are placed on the axis of the first and second differentially pumped vacuum chambers and connected through the second nozzle, as shown in Figure 7. Passing through the third nozzle, ions enter into the third differentially pumped vacuum chamber, where the cylindrical (DC + RF) funnel, with the same central aperture, is placed on the axis (the so-called “RF buncher”). A “weak” viscous low-density gas flow inside the last cylindrical funnel cannot efficiently transport ions and a small DC electric field gradient is additionally applied along this stack of electrodes to drag the ions. This (DC + RF) device allows a “smooth” transfer of the ion beam into a high vacuum. The ions undergo, in the fourth vacuum chamber, a few collisions with helium buffer gas atoms. This may have a negative effect on both the values of an axial energy spread and the transverse emittance of the ion beam, when finally extracted into a vacuum. It is this miniature fourth funnel that helps to dramatically decrease this “last-collisions effect” which is described, e.g., in [41].

Moreover, by applying a small positive capture potential to the last electrode of the cylindrical (DC + RF) funnel, it is possible to accumulate ions inside this device and use it, in addition, as an original, very compact and effective ion beam buncher, as we proposed and numerically investigated for the first time in [42,43].

The geometry of the nozzles and the main design parameters of the described system of extraction RF funnels (see Figure 7) are listed in Table 1 and Table 2, correspondingly. Calculated gas flow rates through the different vacuum chambers and required pumping speeds are listed in Table 3.

RF frequencies and RF amplitudes applied to the extraction RF funnels (see Figure 7) used in Monte Carlo ion-trajectory calculations are listed in Table 4.

To illustrate, Figure 8 shows the results of the gas dynamic simulation for a gas velocity flow field in the system (see Figure 7) of the extraction RF funnels.

The results of the Monte Carlo calculations, listed in Table 5, show the potential of the RF-only funnel technique for ion beam cooling and extraction into a vacuum.

## 4. Laser Ablation Ion Beam Sources

There are three different design variants of laser ablation ion beam sources with the use of gas dynamic RF-only funnel techniques.

### 4.1. Proposal of the Finely Focused Ion Beams for Micro- and Nanoelectronic Technologies

We proposed the first variant in 2008 for the production of finely focused ion beams for micro- and nanoelectronic technologies [41]. Figure 9 shows a schematic of this laser ablation ion beam source. The buffer gas expands into a vacuum through an axial symmetric converging–diverging supersonic nozzle into the RF-only funnel. Through this inner cylinder, which serves as a target holder, a thin rod of target material directly fed into the supersonic buffer gas jet, and a stepper motor (not shown in Figure 9 from Ref. [41]) translates it. The laser beam is focused onto the target rod end by passing through holes in some funnels’ electrodes, as is shown in Figure 9. The details of the nozzle and RF funnels’ geometry, the source operation conditions, as well as the results of the Monte Carlo ion-trajectory simulations can be found in Ref. [41].

Notice that the total gas flow rate through the nozzle (Q_noz_) is 27.8 mbar l/s. So, to maintain 1 mbar pressure in the RF funnel chamber, it is enough to use the vacuum pump at 28 l/s pumping speed. The gas load into the next high vacuum chamber (behind the funnel) (Q_fun_) is 0.06 mbar l/s. It means that ions can be extracted into a vacuum of 2 × 10^−4^ mbar with the use of a relatively small turbomolecular pump of 300 l/s. The ratio (Q_fun_/Q_noz_ = 0.002) clearly shows that this RF-only funnel works as a very effective buffer gas filter, because only a small part of the buffer gas from the nozzle passes through the exit funnel aperture.

To illustrate, Figure 10 and Figure 11 show the results of gas dynamic calculations.

### 4.2. The Laser Ablation Ion Beam Source with the RF-Only Funnel at ETH, Zürich

The second design variant of a laser ablation ion beam source with the RF-only funnel has been proposed [31] and realized [32] at ETH, Zürich. Figure 12 shows the structure of this laser ablation ion beam source setup [32], which was designed for time-of-flight mass spectrometry of the element composition of different samples. It combined a sample placed in a chamber, which filled with a helium buffer gas, a conical converging–diverging nozzle connected to this chamber at short distance behind the sample, a RF-only funnel placed on the nozzle axis in immediate vicinity of the nozzle exit plane and the pulsed laser (frequency-doubled Nd:YAG, 532 nm, 3 ns pulse duration; 7.6 × 10^8^ W power, 10 Hz repetition rate). The laser focused onto the sample surface to a spot diameter of 110 μm, after passing through the funnel and the supersonic nozzle. The primary hot ions generated during ablation were cooled by collisions with buffer gas, accelerated into the subsonic converging part of the nozzle and finally transported through the supersonic diverging part of the nozzle into the RF-only funnel via buffer gas expansion. Then ions were transported by the gas flow into the high vacuum stage, containing the ion optics and mass analyzer.

The funnel consists of 74 electrodes made from 0.1 mm thick stainless steel plates with circular inner apertures. Electrodes of opposite phases are mounted in a crosswise pattern (see the bottom of Figure 12) and separated by 0.7 mm spacers to achieve a 0.3 mm separation of the adjacent electrodes. The first 32 electrodes have an inner aperture of 4.5 mm, which then decreases linearly towards an exit aperture of 0.9 mm. Figure 13 shows a photo of the assembled RF-only funnel.

A schematic view of the converging–diverging nozzle and RF-only funnel setup, combined with the results of the detailed gas dynamic simulation, is shown in Figure 14.

### 4.3. The Laser Ablation Ion Beam Source with the RF-Only Funnel at TU, Darmstadt

We proposed the third design variant of a laser ablation ion beam source in 2016 [42]. Later this proposal was realized at the Technical University (TU), Darmstadt as a prototype of a “He-buffered laser ablation ion source for collinear laser spectroscopy” [33] and it will also be used at the LASPEC experiment at FAIR [44]. 

The setup of this ion beam source [33] consists of three different pressure stages, as shown in Figure 15.

At the first stage, the laser ablation process takes place in the presence of helium. The helium leaks in via a valve through the entrance flange (A) into the ablation unit (B). This unit is located inside the 150-mm main flange (C) with a bore for the laser (D) and a bore for a rotation feedthrough (E). The first and the second pressure stages are connected through a converging–diverging conical nozzle (F). A detailed layout of the setup is shown in Figure 16. 

At the second vacuum stage, the ions are transported by the buffer gas flow through the RF-only funnel (G). Pumping of this chamber is performed by a tube that connects the second stage with the vacuum flange (H) connected to a 40-m^3^/h dry pump (Leybold Ecodry 40 plus). The separation flange (J) has a 1 mm tapered hole in the center, forming the second nozzle (K), which delimits the third vacuum stage. On the other side of this flange, the (DC + RF) funnel is mounted. Flanges (C) and (J) are mounted in series on one arm of a vacuum cross (N) that is equipped with a 900 l/s turbomolecular pump (TMP) to evacuate the remaining gas flow.

A schematic view of the double-funnel system for the laser ablation ion beam source, combined with the results of the gas dynamic simulation for the helium velocity flow field, is shown in Figure 17 [33].

Details of the nozzle and RF funnel geometry and the ion source operation conditions can be found in [33].

Figure 18 shows a typical measured ion signal compared with the result of a Monte Carlo simulation of the total ion transport time. The shape of the simulated signal and its time position are in good agreement with the measured signal.

## 5. “Fair-Wind Gas Cell” Concept

A so-called “fair-wind gas cell” concept was proposed for the first time in [45] for large high-density gas stopping cells. The operation of the fair-wind gas cell was also explored later in follow-up computer experiments and described in [46]. 

The principle schematic of the fair-wind gas cell design is shown in Figure 19. The axial symmetric gas cell consists of the stopping and extraction chamber connected through a central hole. An energetic primary radioactive ion beam enters through an entrance window into the stopping chamber, where the ions are decelerated and thermalized in the buffer gas. 

The idea of a fair-wind gas cell consists of the use of an intense compulsory buffer gas flow through the cell combined with a RF-only funnel. The compulsory gas flow provides fast and highly efficient transportation of the ions to the exit of the stopping chamber. Ions are transported further by the gas flow through the downstream RF-only funnel placed on the axis between the stopping chamber exit hole and the miniature supersonic nozzle. The majority of the flowing-through buffer gas is pumped through gaps between the funnel electrodes. 

Annular slits on the side surfaces of the stopping and extraction chambers (see Figure 19) serve as the buffer gas inlet and outlet, respectively. An intensive buffer gas flow through the cell is maintained by the pressure difference between these slits by pumping. The expensive high-purity buffer gas inlet and outlet may be connected to a clean recycling system (not shown in Figure 19), to keep operation costs within reasonable limits.

The inlet helium pressure (in the stopping chamber) is 1.0 bar; the outlet pressure (in the extraction chamber) is 0.97 bar; the background pressure in the nozzle exhaust chamber is 0.06 mbar. The walls of the fair-wind gas cell are at room temperature. The gas flow rate through the stopping chamber (Q_cell_) is 15 bar l/s; the gas flow rate through the nozzle (Q_noz_) is 121 mbar l/s. The ratio of these flow rates (Q_noz_/Q_cell_ = 0.008) means that less than 1% of the total gas flow through the cell escapes into the vacuum through the nozzle. The main geometric parameters of the stopping and extraction chamber, the RF-only funnel and the nozzle have been presented in [45].

Figure 20 shows the calculated helium velocity flow field in a region of the extraction chamber and the nozzle.

The described “fair-wind gas cell” concept can also be used as an additional option in the UniCell setup [37]. Recently, as proposed in [47], the fair-wind gas cell’s design was optimized for the parameters of the super heavy ion beam extracted from the recoil separator TASCA [40]. 

A schematic of the UniCell setup variant combined with the fair-wind gas cell for use in SHE chemistry shown in Figure 21. The SHE ions from the recoil separator are introduced through a thin entrance window (1) into the stopping chamber of the gas cell flushed with the pure helium at a pressure (P_0_) of 1 bar. The thermalization of SHE ions with the buffer gas occurs inside the cylindrical part of the tube (2), which has a length of 60 mm and an inner diameter of 70 mm (the same sizes as in the gas cell described in [37]). The subsonic laminar compulsory gas flow through the entrance chamber is maintained by the gas recirculation under the pressure difference (P_0_ − P_1_) of 50 mbar, which allows the effective transport of ions to the RF-only funnel (3) placed on the axis in the extraction chamber. This funnel consists of a stack of 0.1 mm-thick metal electrodes separated by 0.25 mm gaps. The design of the funnel’s electrodes is the same as used in [32,33]. 153 electrodes are assembled on four supporting rods. The subsonic gas flow inside the funnel effectively transports the ions to the exit nozzle, whereas the RF field confines the ions inside the funnel by repelling them from the funnel surface. The throat diameter, exit diameter and length of the diverging conical exit nozzle (4) (see Figure 21) are 0.3 mm, 4 mm and 7.4 mm, respectively.

The gas flow unfoldes near the entrance window and flows towards the exit hole, as shown in Figure 22. The calculated gas flow velocity along the axis of the stopping chamber, shown in Figure 23, is nearly constant in the cylindrical part (approximately a few m/s) and increases in the conical part up to 173 m/s.

The extraction efficiency for ions with a mass of 290 a.u. through the nozzle (marked (4) in Figure 21) as a function of the RF voltage amplitude applied to the funnel electrodes was obtained in Monte Carlo simulations. The results are shown in Figure 24 for a voltage range from 60 to 100 V. As can be seen in Figure 24, a moderate RF voltage amplitude (peak to peak) (V_pp_) of 100 at the RF frequency of 5 MHz is enough to effectively extract the ions from the fair-wind gas cell.

## 6. Windowless Gas Dynamic Ion Beam Cooler and Buncher

We suggested an idea of the cooler for intense low-energy ion beams with the RF-only funnel placed on the beam axis downstream of the supersonic nozzle exit plane for the first time in 2003 [48]. After several years of RF-only funnel technique development, we have developed a new concept of a windowless gas dynamic ion beams cooler and buncher. This windowless gas dynamic ion beam cooler/buncher, as described in detail in [49], has a very simple and ultra-compact design, the schematic view of which, combined with the results of the detailed gas dynamic simulation for the helium velocity flow field, is shown in three figures (Figure 25, Figure 26 and Figure 27). We deliberately divided the total schematic view of this setup into three parts to show in more detail the complex shock wave structure of a supersonic helium jet inside the first two RF-only funnels, as well as gas flow details in the area of the first nozzle inlet and in the third (RF + DC) funnel, which we have called the RF buncher. 

Through the inner tube placed inside the first nozzle, a primary energetic ion beam is directly injected into the expanded supersonic buffer gas jet and then enters into the first RF-only funnel, as shown in Figure 25.

The first RF-only funnel, placed on the axis at 13.65 mm downstream from the first nozzle exit plane, allows for the primary energetic ions’ deceleration down to the gas jet velocity, and their cooling down to the static temperature of the supersonic jet. Moreover, the buffer gas flow transports ions effectively and quickly to the second conical diverging nozzle. We can say that in a sense, this first RF-only funnel plays the role of a windowless gas stopping cell. The gap value between the first nozzle and the first RF-only funnel is not a critical one, and it can be larger, or smaller (even zero) because the gas flowing from the first nozzle completely flows into the first funnel (see Figure 25 and Figure 26).

Passing through the second nozzle, which provides conditions for differential pumping, the ions are transported by the gas flow through the second RF-only funnel to the third miniature diverging nozzle. After that, the ions move into the third differentially pumped vacuum chamber, where the cylindrical (DC + RF) funnel is placed on the axis in immediate vicinity behind the miniature third nozzle. It is important to note here that the third funnel (see Figure 27) can be also effectively used as the ultra-compact (only 18 mm in length) RF buncher, as well. 

The main design parameters of the nozzles and funnels, as well as the gas flow rates through different vacuum chambers and required pumping speeds are listed in the Tables of Ref. [49]. The design of the RF funnel’s electrodes and their assembly into the funnel are similar to those described in [33]. 

Notice that, for the proper operation of the previously proposed ion cooler [48], the huge pumping capacity of the turbo-molecular pumps, of 5400 l/s (108 mbar l/s gas flow rate through the nozzle at 2 × 10^−2^ mbar background pressure), needs to be used. This very big disadvantage of this previous concept of a cooler [48] can be explained by the fact that in [48] there are no conditions for differential pumping between the RF-only funnel and the RFQ installed downstream (see Figure 3). 

Moreover, at the background pressure of 2 × 10^−2^ mbar in the cooler [48], the mean free path length of the ions is only approximately 10 mm. It means that the ions, which should be accelerated by DC voltage applied between the exit of the RFQ and the skimmer for their focusing and final extraction into the high vacuum, will undergo the last collisions with background atoms. As a result, it will lead to both the loss of part of the ion beam and its heating (i.e., an increase in the extracted beam emittance).

The buffer gas flowing around the inner tube in the first nozzle (see Figure 27) creates a zone of strong rarefaction behind its end. In this case, we can say that the first nozzle works as an efficient ejector pump. This effect is illustrated in Figure 28, where the results of the gas dynamic simulation for the gas density distribution along the axis are shown for the region of the first nozzle. As can be seen in Figure 28, at a distance of 10 mm from the inlet of the first nozzle’s inner tube, the gas density drops by 4 orders of magnitude. At the same time, the corresponding average gas pressure (that is equivalent to the gas density at room temperature) inside the first funnel is approximately 24 mbar. That is why the thickness of the gas stream (in the region between the exit of the first nozzle and the entrance to the second nozzle) is approximately 6.6 × 10^18^ atoms/cm^2^. It is another large and important advantage of the ion beam cooler [49] over the previous proposal [48] because here the buffer gas thickness is 22 times higher than the thickness of the gas jet in [48].

The ion beam cooler/buncher, as presented in Ref. [49], can be effectively used for the direct gas dynamic cooling of the ion beams, with primary energies up to 300 keV. It means, in particular, that for this cooler [49] there is no need for any high-voltage platform, which is mandatory for most of the existing RFQ ion coolers.

The results of Monte Carlo ion-trajectory simulations for the total transmission efficiency of the ions of different masses and different primary energies through the cooler and buncher [49] are presented in Figure 29.

The calculated transport time for distributions of ions with a mass of 100 through the ion beam cooler/buncher for different primary ion energies is shown in Figure 30. As can be seen here, the transport time is independent of primary energy. However, the width of the time distributions increases with the primary ion energy. 

The results of the Monte Carlo calculations for the continuous and bunched ion beams for different ion masses at a gas stagnation pressure (P_0_) of 250 mbar are listed in Table 6 and Table 7, correspondingly. The presented values correspond to a 90% fraction of the extracted ion beams.

To trap ions inside the RF buncher, +2.0 V additional potential is applied to the last RF buncher electrode during the accumulation time of 1 ms. 

Just for comparison, if the normalized transverse emittance value of 170 π∙mm∙mrad∙[eV]^1/2^ is scaledfor an energy of 60 keV (the continuous ISOLDE ion beam energy [17,18]), the calculated emittance value (for 90% level) becomes 0.69 π∙mm∙mrad, which looks very promising.

To illustrate the data in Table 7, the time structure and velocity distribution of the extracted pulsed beam with an ion mass of 80 are shown in Figure 31 and Figure 32, correspondingly.

## 7. Summary

The review of the development and present status of a novel gas dynamic RF-only funnel technique for the low-energy ion beam cooling and extraction into a vacuum is presented.

It is shown that this simple, original and efficient technique makes possible the production of high-quality continuous and pulsed low-energy ion beams with a wide mass range and with a very small transverse and longitudinal emittance.

## Figures and Tables

**Figure 1 micromachines-14-01771-f001:**
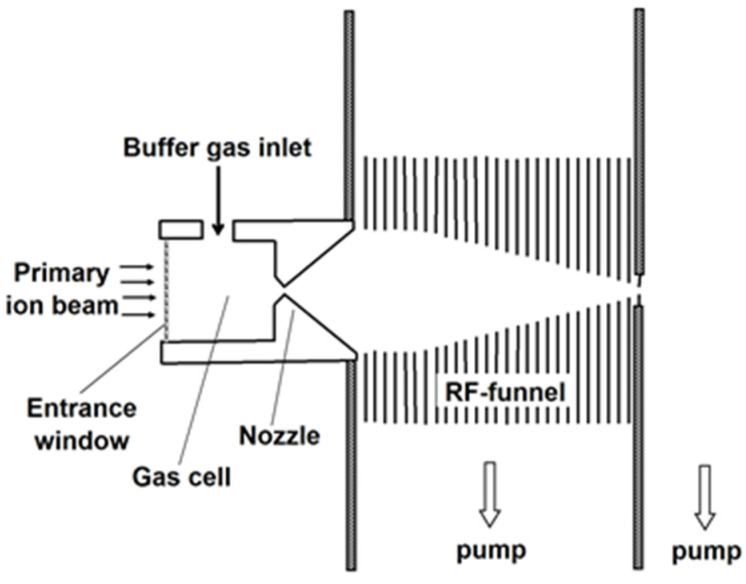
The schematic of the new approach to the extraction system design with the use of RF-only ion funnel. Results of gas dynamic simulations for the case of LEBIT project at MSU shown in slides 6–10 of our presentation [26] to illustrate this approach’s operation.

**Figure 2 micromachines-14-01771-f002:**
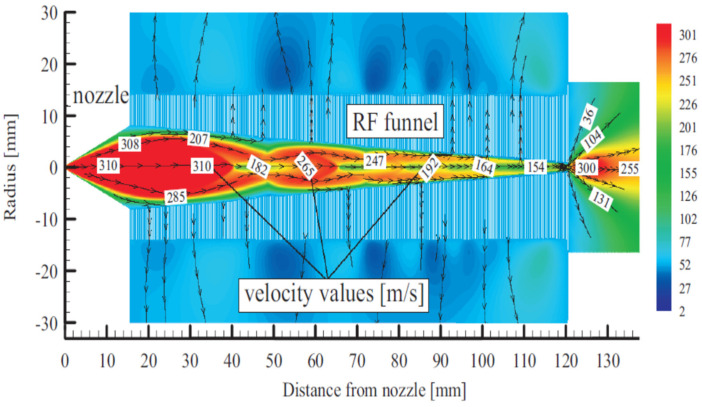
Schematic view of the RF-only funnel combined with results of gas dynamic simulation for xenon velocity flow field. The stagnation input gas pressure and temperature are PA = 10 bar and TA = 300 K, correspondingly. The gas pressure in the RF-only funnel chamber is PB = 8 × 10^−3^ mbar and the background gas pressure in the vacuum chamber behind the funnel exit is PC = 1.5 × 10^−3^ mbar. Black arrowed lines show the gas flow direction. The color scale indicates velocity in m/s [30].

**Figure 3 micromachines-14-01771-f003:**
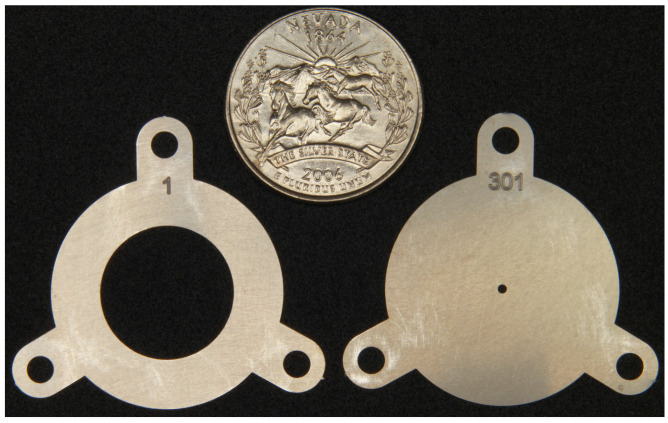
Picture of the first (ID 16.0 mm) and last electrodes (ID 1.0 mm). A US quarter (OD 24.3 mm) shown for scale [30].

**Figure 4 micromachines-14-01771-f004:**
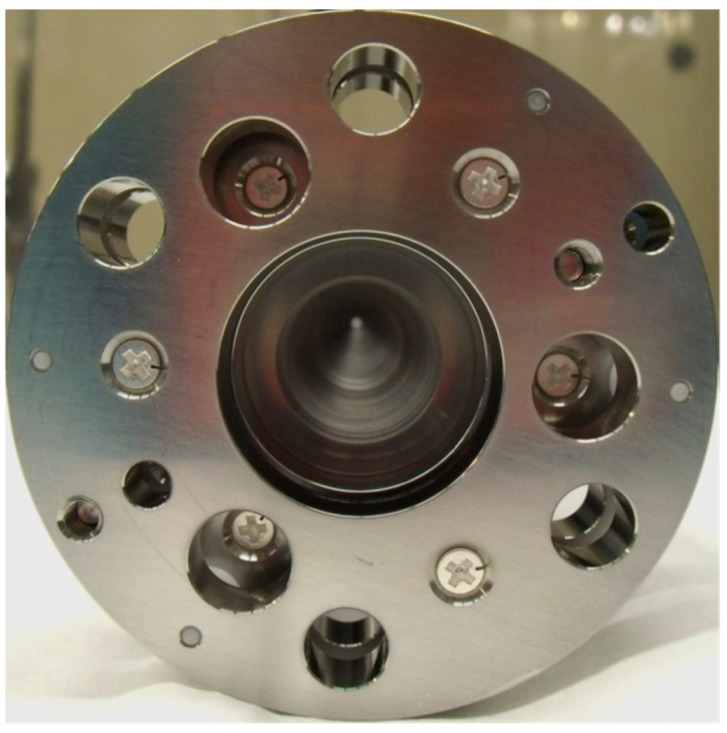
Picture of RF-only funnel assembly prior to installation on the nozzle, view from base to exit. The conical cavity ending with diameter 1.0 mm (ID of electrode #301) is visible. Figure 5 shows this assembly (left) mounted to the nozzle (right) [30].

**Figure 5 micromachines-14-01771-f005:**
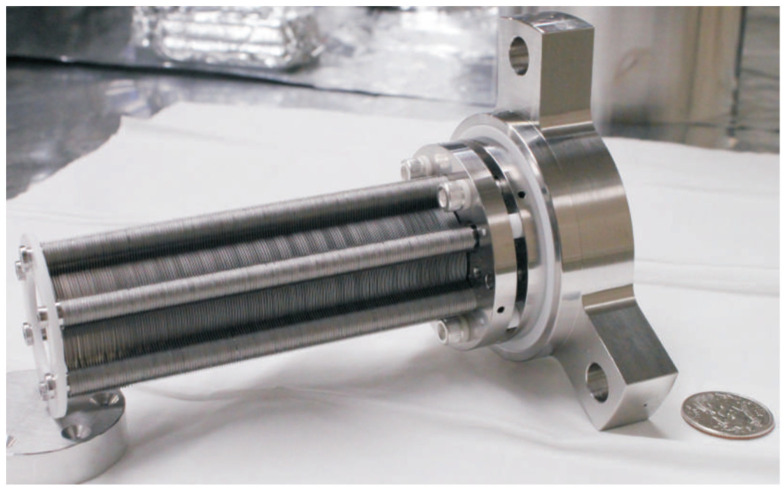
Picture of assembled RF-only funnel mounted on the nozzle flange. Gas is injected from right [30].

**Figure 6 micromachines-14-01771-f006:**
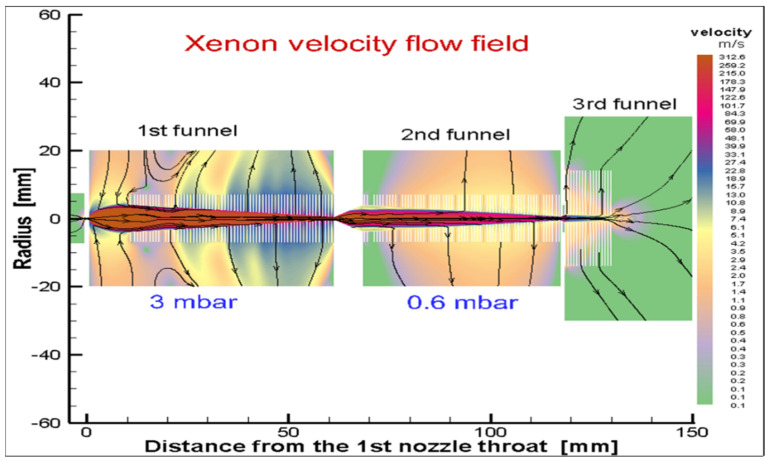
Schematic view of the triple-funnel system combined with results of detailed gas dynamic simulation. The stagnation input gas pressure and temperature are Po = 10 bar and To = 300 K, correspondingly. Black arrowed lines show the gas flow direction [38].

**Figure 7 micromachines-14-01771-f007:**
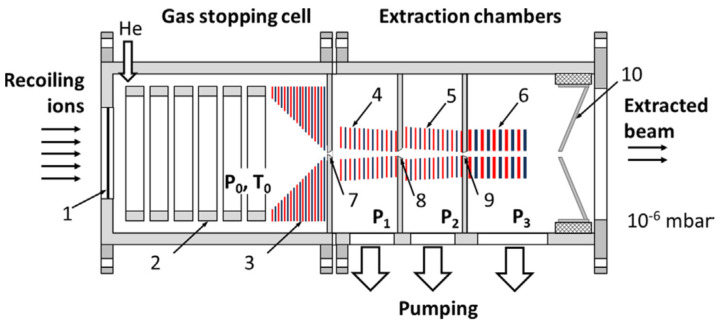
The schematic of the new universal high-density gas stopping cell setup for study of nuclear properties. Recoiling ions from a kinematic recoil separator are introduced into the gas stopping cell through a thin window (1). The DC cage (2) and (DC + RF) funnel (3) are mounted inside the gas stopping cell. P_0_ and T_0_ are stagnation helium pressure and temperature, correspondingly. The RF-only funnels (4), (5) and a RF buncher (6) are placed downstream the exit of funnel (3) in the next differentially pumped chambers, which have the background gas pressures P_1_, P_2_, P_3_, correspondingly. The 1st, 2nd and 3rd diverging conical nozzles are shown as (7), (8) and (9). The conical extraction electrode (10) serves for electrostatic acceleration and final extraction the ion beam into high vacuum [37].

**Figure 8 micromachines-14-01771-f008:**
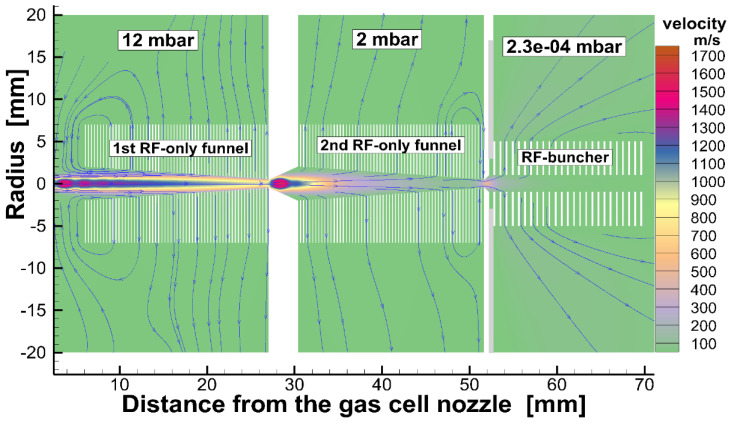
Results of the gas dynamic simulation for helium gas velocity flow field in the system of extraction RF funnels shown in Figure 7. The stagnation pressure and temperature in the gas stopping cell are 1.0 bar and 300 K, correspondingly. Blue arrowed lines show the gas flow directions. The extraction conical electrode (#10 in the Figure 7) placed on the axis behind the RF buncher exit not shown [37].

**Figure 9 micromachines-14-01771-f009:**
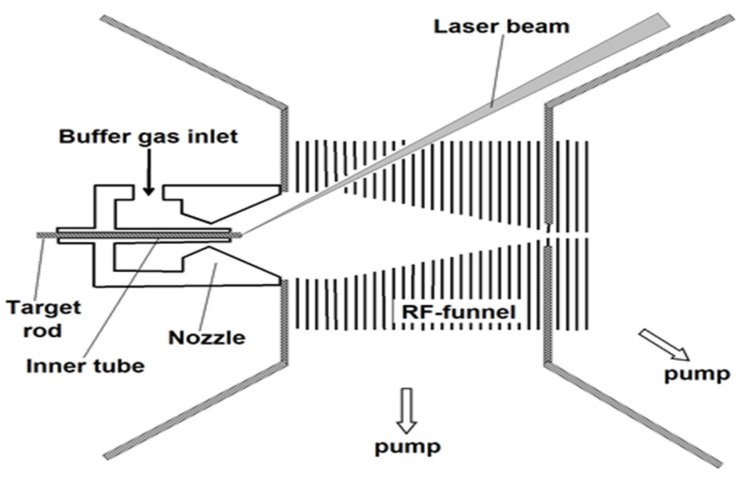
The schematic view of the ablation ion beam source for focused ion beam production [41].

**Figure 10 micromachines-14-01771-f010:**
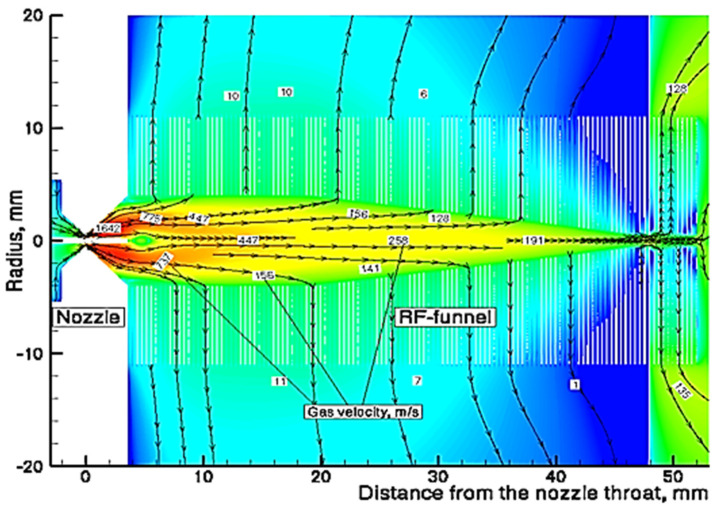
Results of the gas dynamic simulation for helium velocity flow field: black arrowed lines are the helium streamlines; the red color represents maximum and the blue color minimum velocity values [41].

**Figure 11 micromachines-14-01771-f011:**
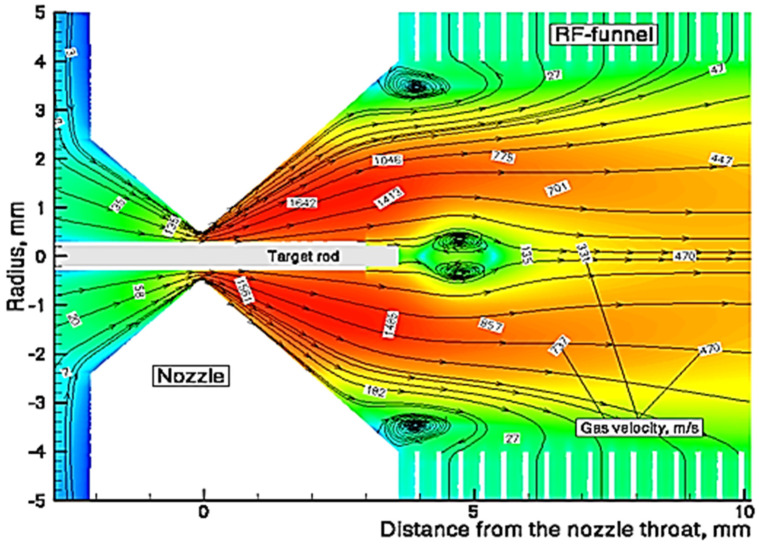
Results of the gas dynamic simulation for helium velocity flow field in the area of the nozzle: black arrowed lines are the helium streamlines; the red color represents maximum and the blue color minimum velocity values [41].

**Figure 12 micromachines-14-01771-f012:**
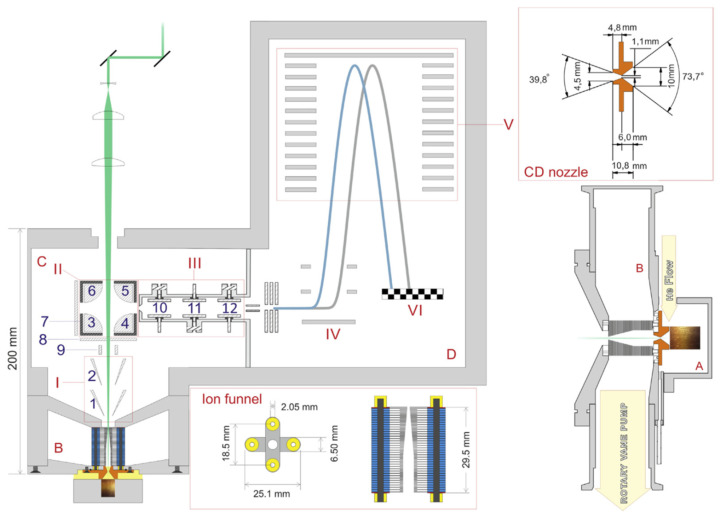
(I) Extraction cones, (II) quadrupole deflector, (III) einzel lens, (IV) extraction lens, (V) reflectron, (VI) MCP detector. Pressure stages: (**A**) ablation region, (**B**) funnel chamber, 3 mbar, (**C**) ion optic region, 10^−4^ mbar, (**D**) TOFMS, 10^−6^ mbar. Labels 1 to 12 indicate individual ion optic elements; further explanation can be found in the text of Ref. [32]. Top-right: close-up and dimensions of the converging–diverging nozzle. Bottom right: details of the sample–nozzle–funnel arrangement [32].

**Figure 13 micromachines-14-01771-f013:**
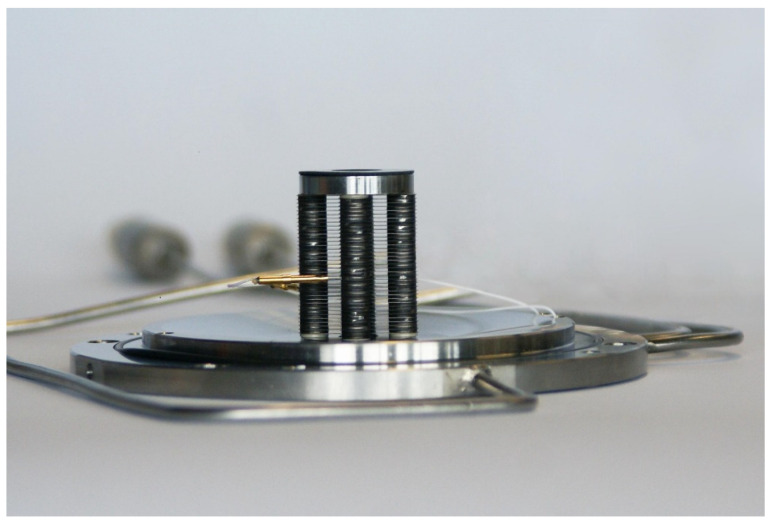
Picture of the assembled RF-only funnel.

**Figure 14 micromachines-14-01771-f014:**
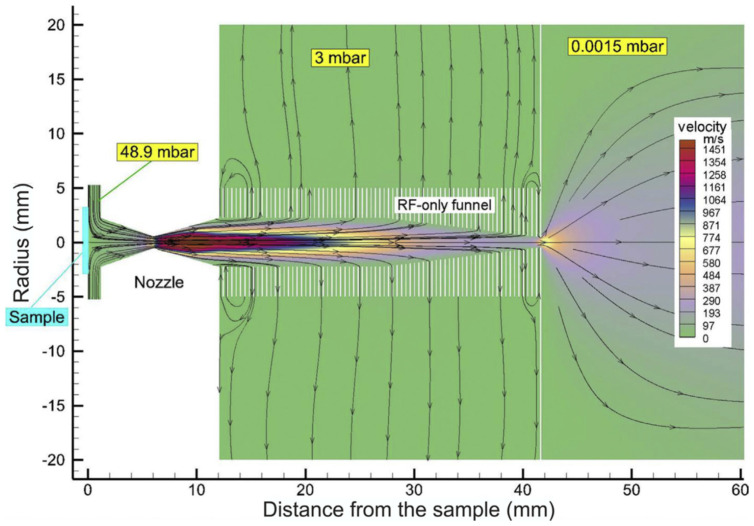
Schematic view of the converging–diverging nozzle and RF-only funnel setup combined with results of the detailed gas dynamic simulation [32].

**Figure 15 micromachines-14-01771-f015:**
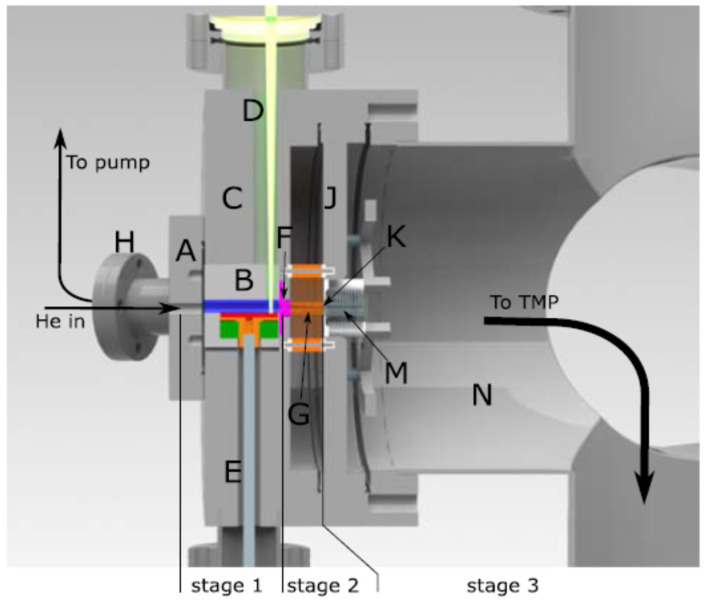
Overview of the setup with laser beam (D), ablation stage (B), funnel stage (F,G) and detection stage (K,M in N). Details provided in the text of Ref. [33] and it presented below, as well.

**Figure 16 micromachines-14-01771-f016:**
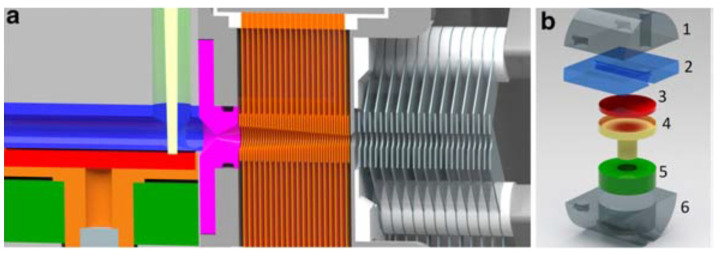
Detailed layout of the setup: (**a**) technical drawing of the ion source with ablation unit (color code as in the figure to the right): 1st nozzle (magenta), RF-only funnel section (orange), and (DC + RF) funnel (gray) with insulators (white). (**b**) Exploded view of the ablation unit: (1) top part with laser bore, (2) middle part with 5 mm ablation chamber, (3) 1 inch sputter target, (4) target holder, (5) ceramic bearing, (6) bottom part [33].

**Figure 17 micromachines-14-01771-f017:**
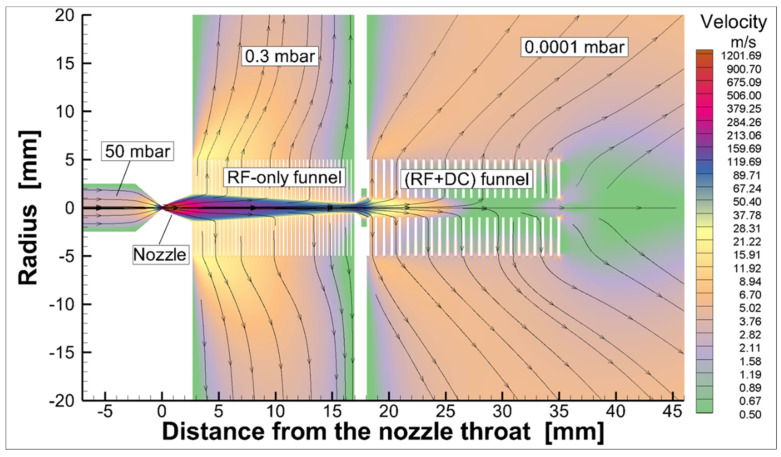
Schematic view of the laser ablation ion beam source system combined with results of the gas dynamic simulation for helium velocity flow field. Black arrowed lines show the gas flow direction. The 1st and 2nd nozzle throats’ diameters are 0.3 mm and 1.0 mm, correspondingly. The pressure in the ablation chamber is 50 mbar, resulting in a pressure of 0.3 mbar in the chamber of the RF-only funnel and 10^−4^ mbar in the chamber of the (RF + DC) funnel [33].

**Figure 18 micromachines-14-01771-f018:**
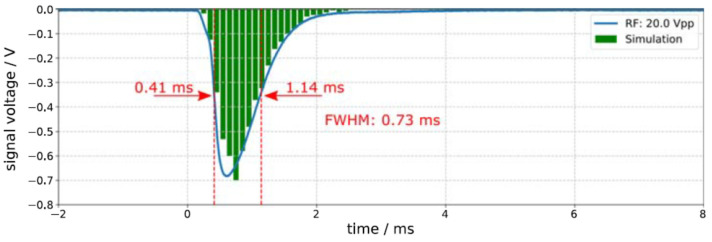
Typical measured ion signal compared with the result of Monte Carlo simulation of the total ion transport time. The data acquisition was triggered by the signal for the Q-switch of the pulsed laser, which is marked as zero on the horizontal axis.

**Figure 19 micromachines-14-01771-f019:**
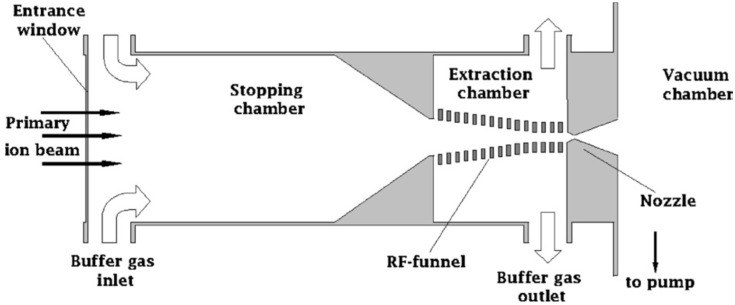
Principle schematic of the fair-wind gas cell concept [45].

**Figure 20 micromachines-14-01771-f020:**
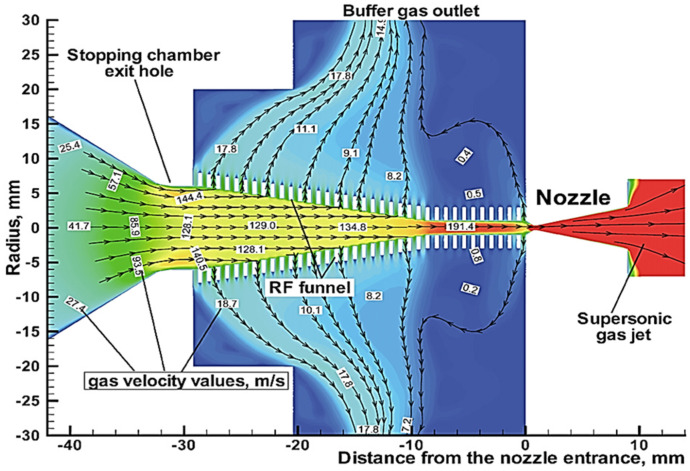
The calculated helium velocity flow field in a region of the extraction chamber and the nozzle [45].

**Figure 21 micromachines-14-01771-f021:**
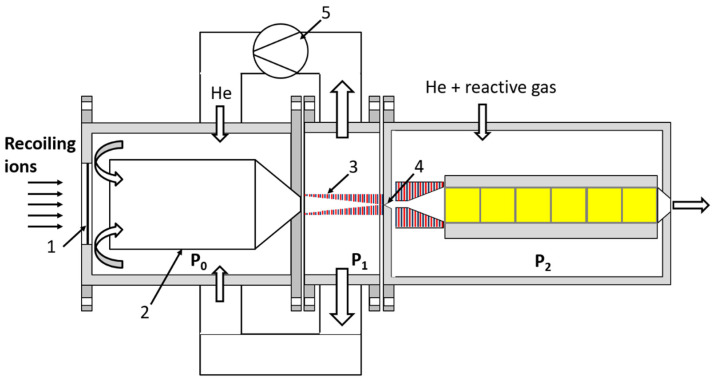
Schematic of the UniCell setup variant for use in SHE chemistry combined with the fair-wind gas cell (see Figure 7 above for comparison). Recoiling ions from a kinematic recoil separator TASCA [40] are introduced into the stopping chamber through a thin entrance window (1). The tube (2) placed at 10 mm distance from the entrance window has a total length of 95 mm (60 mm of cylindrical part with the inner diameter of 70 mm and rounded cone-shaped part of 35 mm length and an exit diameter of 8 mm). The RF-only funnel (3) in the extraction chamber of this gas cell, with a length of 55.55 mm, consists of 153 thin metal electrodes separated by gaps. The diverging conical nozzle (4), with a throat diameter of 0.3 mm, is placed on the axis in immediate vicinity of the funnel exit plane. The inlet and outlet pressures, P_0_ and P_1_, are equal to 1.0 bar and 0.95 bar, correspondingly. The pump (5) causes the gas circulation. The connection to a chemistry detector via an ejector [47] is shown on the right side of the figure. The pressure in the “chemistry chamber” (P_2_) = 0.9 bar [47].

**Figure 22 micromachines-14-01771-f022:**
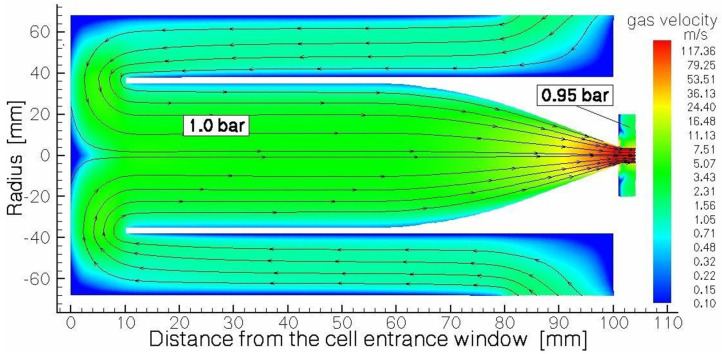
Results of the gas dynamic simulation for helium gas flow field in the stopping chamber of the fair-wind gas cell. Black arrowed lines show the gas flow direction. Note that the vertical and horizontal scales are different [47].

**Figure 23 micromachines-14-01771-f023:**
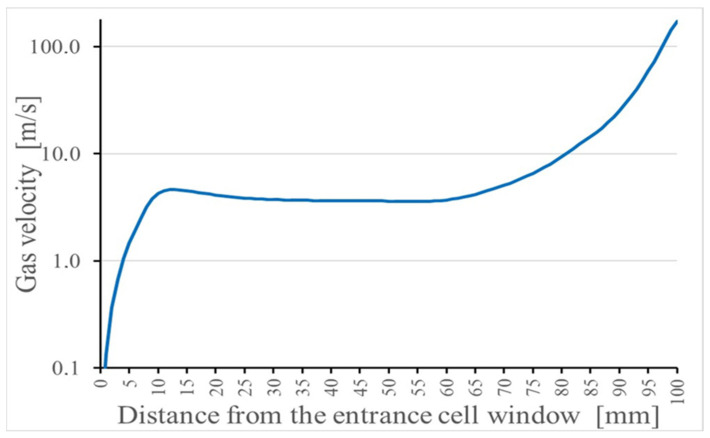
Results of the gas dynamic simulation for helium gas velocity distribution along the axis in the stopping chamber of the gas cell. The inlet and outlet pressures, P_0_ and P_1_, are equal to 1.0 bar and 0.95 bar, correspondingly [47].

**Figure 24 micromachines-14-01771-f024:**
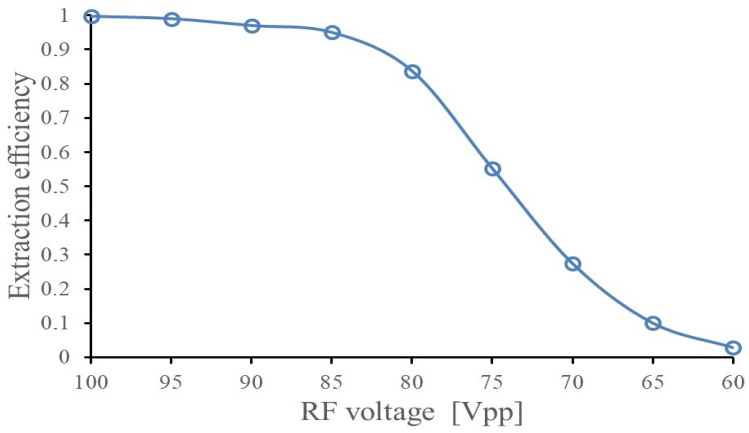
Results of the Monte Carlo ion-trajectory simulations for the extraction efficiency of SHE ions through the nozzle (marked (4) in Figure 21) as a function of applied RF voltage (peak-to-peak). Ion mass is 290, RF frequency is 5 MHz. The inlet and outlet pressures, P_0_ and P_1_, are equal to 1.0 bar and 0.95 bar, correspondingly [47].

**Figure 25 micromachines-14-01771-f025:**
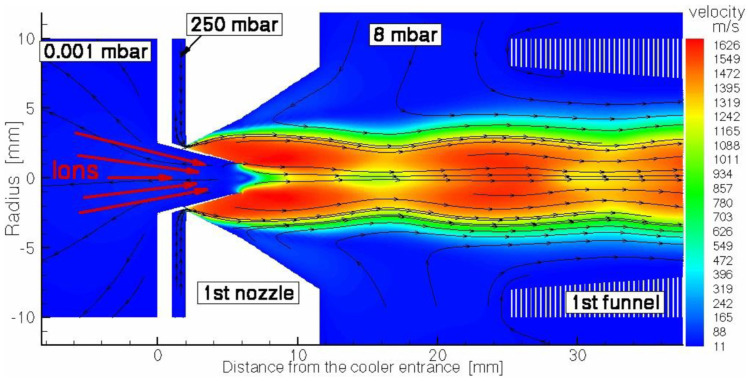
Schematic view of the first entrance part of the ion cooler (the region of the 1st nozzle having a central conical tube for the primary ion beam injection) combined with results of the gas dynamic simulation for helium velocity flow field. The stagnation input gas pressure and temperature are 250 mbar bar and 300 K, correspondingly. The pressure in the chamber of 1st RF-only funnel (downstream of the 1st nozzle exit plane) is 8 mbar, the background pressure in the vacuum chamber in front of the 1st nozzle (left part of the figure) is 1.4∙× 10^−4^ mbar. Black arrowed lines show the gas flow direction. For details of the nozzle geometry, see in the text of Ref. [49].

**Figure 26 micromachines-14-01771-f026:**
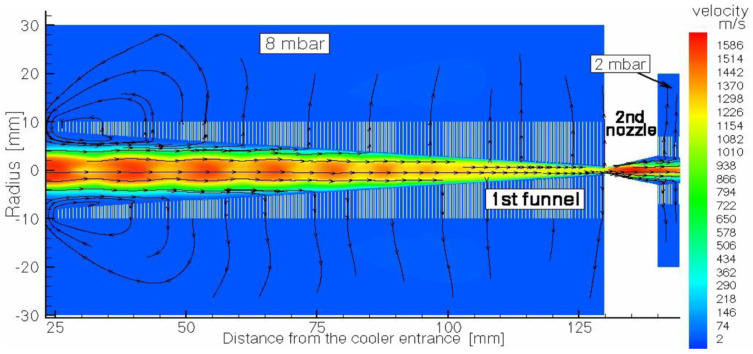
Schematic view of the second (middle) part of the ion beam cooler (the region of the 1st RF-only funnel) combined with results of the gas dynamic simulation for helium velocity flow field. The background gas pressure of 8 mbar in the vacuum chamber is maintained by pumping. The 1st RF-only funnel is connected to the next 2nd RF-only funnel through the conical diverging 2nd nozzle, which allows for the possibility of differential pumping. Black arrowed lines show the gas flow direction [49].

**Figure 27 micromachines-14-01771-f027:**
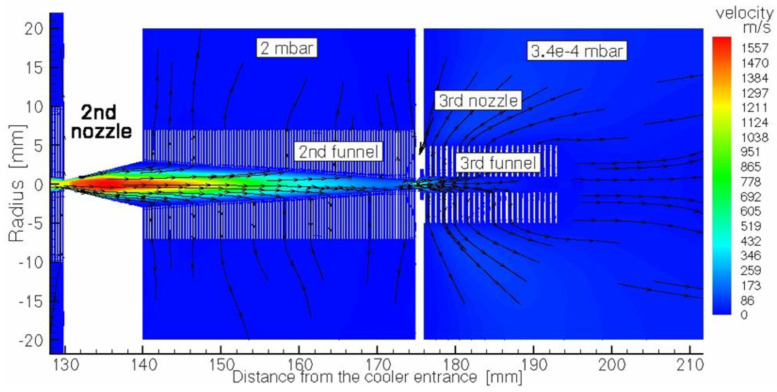
Schematic view of the third (end) part of the ion beam cooler combined with results of the gas dynamic simulation for helium velocity flow field. It consists of two differentially pumped vacuum chambers. The 2nd RF-only funnel in the left chamber is connected to the 3rd (RF + DC) funnel (or RF buncher) placed in the next chamber (on the right) through the 3rd short conical diverging nozzle. Background pressures of 2 mbar and 3.4 × 10^−4^ mbar in both vacuum chambers are maintained by pumping. The extraction electrode placed on the axis behind the RF buncher exit is not shown. Black arrowed lines show the gas flow direction [49].

**Figure 28 micromachines-14-01771-f028:**
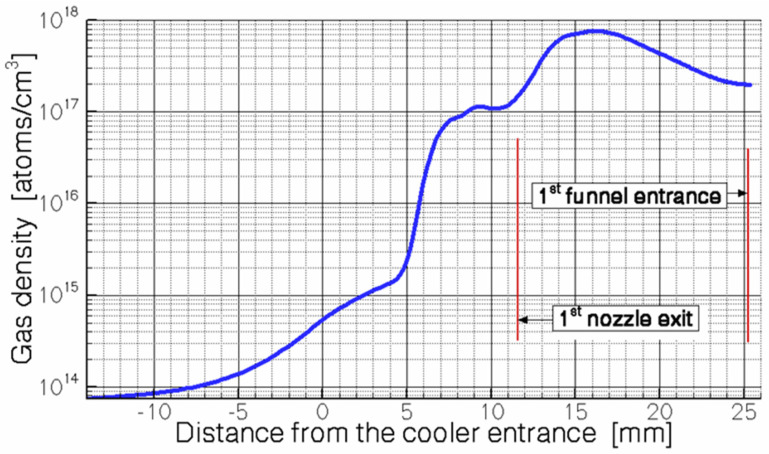
Calculated helium density distribution along the axis. The gas stagnation pressure (P_0_) = 250 mbar [49].

**Figure 29 micromachines-14-01771-f029:**
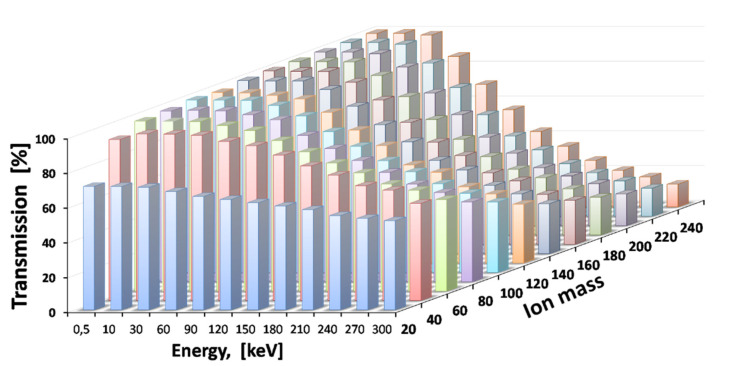
Results of the Monte Carlo ion-trajectory simulations for total transmission efficiency through the cooler and buncher for ions of different masses and different primary energies. The gas stagnation pressure (P_0_) = 250 mbar [49].

**Figure 30 micromachines-14-01771-f030:**
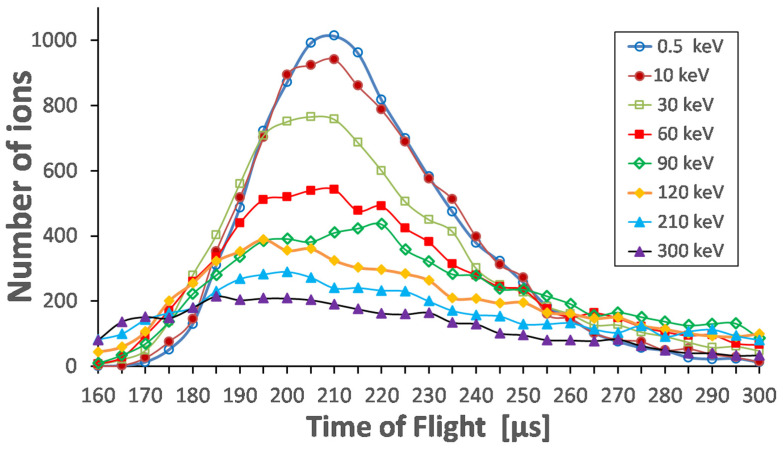
Results of Monte Carlo ion-trajectory simulations for the transport time of ions with mass of 100 through the ion beam cooler and buncher for different primary ion energies. The gas stagnation pressure (P_0_) = 250 mbar. Total number of calculated ions for each energy is 10,000 [49].

**Figure 31 micromachines-14-01771-f031:**
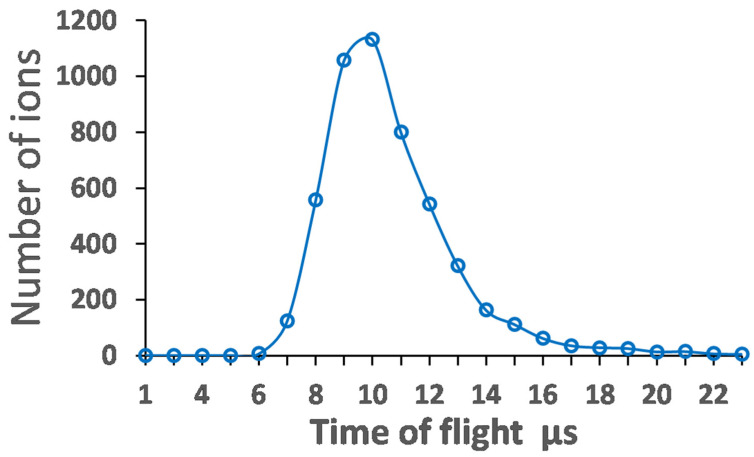
Results of the Monte Carlo ion-trajectory simulations for the time structure of the extracted pulsed beam with ion mass of 80. RF amplitude (peak-to-peak) applied to the RF buncher electrodes (V_pp_) = 85, RF frequency = 10 MHz. Extraction electrode placed at 10 mm downstream of the RF buncher exit at potential = −100 V. Number of extracted ions is 5000 [49].

**Figure 32 micromachines-14-01771-f032:**
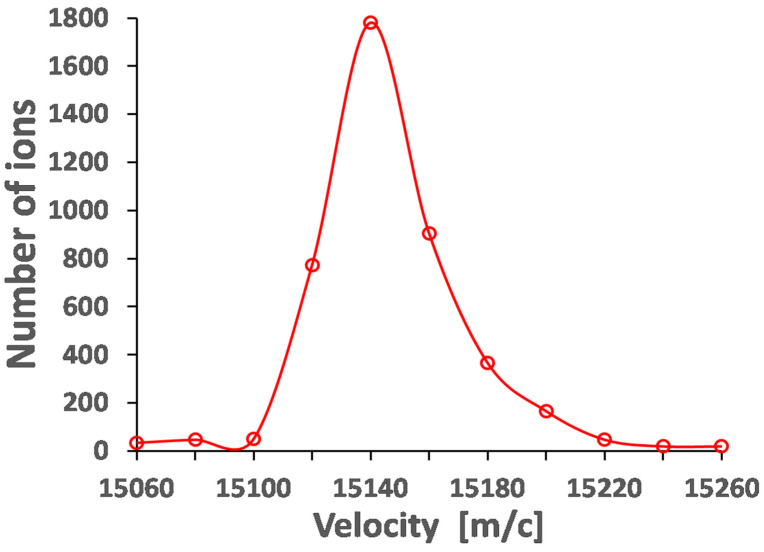
Results of the Monte Carlo ion-trajectory simulations for the longitudinal velocity distribution of the extracted pulsed beam with ion mass of 80. RF amplitude (peak-to-peak) applied to the RF buncher electrodes (V_pp_) = 85, RF frequency = 10 MHz. Extraction electrode placed at 10 mm downstream of the RF buncher exit at potential = −100 V. Number of extracted ions is 5000 [49].

**Table 1 micromachines-14-01771-t001:** Geometry of the nozzles [37].

Nozzle	1st	2nd	3rd
Half-angle of diverging cone	26.5°	26.5°	45°
Length of diverging cone	3.7 mm	3.0 mm	0.5 mm
Throat diameter	0.3 mm	1.0 mm	1.0 mm
Exit diameter	4.0 mm	4.0 mm	2.0 mm
Gap between nozzle and funnel	5.0 mm	-	0.5 mm

**Table 2 micromachines-14-01771-t002:** Main design parameters of the extraction RF funnels system [37].

Funnel	1st and 2nd (RF-Only)	Buncher (RF + DC)
Entrance aperture diameter	4.0 mm	2.0 mm
Exit aperture diameter	1.0 mm	2.0 mm
Electrode thickness	0.1 mm	0.2 mm
Inter-electrode spacing	0.25 mm	0.5 mm
Number of electrodes	60	25

**Table 3 micromachines-14-01771-t003:** Calculated gas flow rates through the different vacuum chambers and required pumping speeds when stagnation pressure and temperature in the gas stopping cell are 1.0 bar and 300 K, correspondingly [37].

Parameter	Gas Cell	1st RF-Oonly Funnel	2nd RF-Only Funnel	RF Buncher
He gas pressure	1.0 bar	12 mbar	2 mbar	2 × 10^−4^ mbar
Gas flow rate through chamber (mbar l/s)	27.0	20.57	6.2	0.23
Required pumping speed (l/s)	-	1.71	3.1	1000

**Table 4 micromachines-14-01771-t004:** RF frequencies and RF amplitudes applied to the extraction RF funnels. Extraction electrode potential = −100 V [37].

	1st RF-Only Funnel	2nd RF-Only Funnel	RF Buncher
RF amplitude (V_pp_)	15	15	80
RF frequency (MHz)	5	5	5
3rd nozzle-buncher DC bias	-	-	−0.35 V
DC potential gradient	-	-	−0.035 V

**Table 5 micromachines-14-01771-t005:** Results of calculations for the extracted continuous ion beams. Extraction electrode potential = −100 V, gas cell temperature (T_0_) = 300 K, stagnation helium pressure (P_0_) = 1 bar, RF frequency = 5 MHz [37].

Ion Mass	290	100	27
Total transmission efficiency (%)	92.9 ± 2.5	89.6 ± 2.4	84.2 ± 2.3
Longitudinal velocity (m/c) energy (eV)	7990 95.8	13,580 95.47	26,150 95.5
Longitudinal (90%) velocity spread (m/c) energy spread (eV)	90 0.02416	230 0.0544	250 0.0171
Radial velocity (m/c) energy (eV)	230 0.079	360 0.0797	850 0.141
Radial velocity spread (m/c) energy spread (eV)	340 0.344	650 0.434	1250 0.434
Beam radius (90%) (mm)	0.95	0.85	0.89
Transverse emittance ε_x,y_ (π∙mm∙mrad)	19.34	16.54	20.46
Normalized emittance ε^N^_x,y_ = ε_x,y_∙[E]^1/2^ (π∙mm∙mrad [eV]^1/2^)	189.27	161.63	199.9
Time of flight (µs)	200	200	200
Time of flight width ΔT (µs)	127	71	75

**Table 6 micromachines-14-01771-t006:** Results of calculations for the extracted continuous ion beams. Extraction electrode potential = −100 V [49].

Ion Mass	20	40	80	120	160	200	240
Total transmission efficiency (%)	98.3	99.1	99.6	99.5	99.6	99.7	99.4
Longitudinal (90%) energy spread (eV)	1.15	0.28	0.17	0.11	0.09	0.07	0.06
Transverse (90%) energy spread (eV)	0.65	0.26	0.31	0.32	0.33	0.33	0.35
Normalized transverse emittance (90%) (π∙mm∙mrad∙[eV]^1/2^)	151.7	171.6	159.8	182	183.6	183.8	164.7
RF amplitude (peak-to-peak) (V_pp_)	60	70	85	105	120	140	150

**Table 7 micromachines-14-01771-t007:** Results of simulations for the extracted bunched ion beams. Extraction electrode potential = −100 V [49].

Ion Mass	20	40	80	120	160	200	240
Total transmission efficiency (%)	96	98	99.7	99	98.6	99.2	98.9
Longitudinal (90%) energy spread (eV)	0.47	0.33	0.15	0.06	0.05	0.05	0.04
Bunch time (90%) width (µs)	10	6.2	6.8	8	9.6	12	13
Longitudinal emittance (90%) (eV µs)	4.7	2.05	1.04	0.48	0.51	0.64	0.55
Transverse (90%) energy spread (eV)	0.65	0.26	0.31	0.32	0.33	0.33	0.35
Normalized transverse emittance (90%) (π∙mm∙mrad∙[eV]^1/2^)	151.7	171.6	159.8	182	183.6	183.8	164.7
RF-Amplitude (peak-to-peak) (V_pp_)	60	70	85	105	120	140	150

## Data Availability

The data presented in this study are available upon request from the corresponding author.

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
