# Peer review of "(untitled)"

_micromachines, 2023, doi:10.3390/mi14091771_

Round 1

Reviewer 1 Report

I would have liked to see the gas jet in this figure 1.  There is a sentence in the General Description section,  I would also have liked to see how these energy spreads and emittances, for this ion source compare to conventional sources, e.g. RF, duoplasmatrons, liquid metal ion sources, ...

There are a lot of examples where a word is in the wrong position of the sentence and the improper use or omission of articles in this paper.   But this is not affecting the readability and I suggest just making the edits I have indicated in comments in the MS.

Author Response

Many thanks to Reviewer 1

Reviewer 2 Report

The article is devoted to an overview of the modern techniques for ion beams extraction into vacuum. A detailed analysis of the design of the elements of the RF-only funnels was carried out and Monte-Carlo simulations of the gas flow was carried out. I recommend to publish this manuscript in its present form after the check of some misspellings, i.e. "vaccum" on p.6, "blation" on p.13, "Figur 22" and others.

Author Response

Many thanks to Reviewer 2
